# DOMAIN ADAPTATION WITH ASYMMETRICALLY-RELAXED DISTRIBUTION ALIGNMENT

**Yifan Wu**
Carnegie Mellon University
yw4@cs.cmu.edu

**Ezra Winston**
Carnegie Mellon University
ewinston@cs.cmu.edu

**Divyansh Kaushik**
Carnegie Mellon University
dkaushik@cs.cmu.edu

**Zachary Lipton**
Carnegie Mellon University
zlipton@cmu.edu

## ABSTRACT

Domain adaptation addresses the common problem when the *target* distribution generating our test data drifts from the *source* (training) distribution. While absent assumptions, domain adaptation is impossible, strict conditions, e.g. *covariate* or *label* shift, enable principled algorithms. Recently-proposed domain-adversarial approaches consist of aligning source and target encodings, often motivating this approach as minimizing two (of three) terms in a theoretical bound on target error. Unfortunately, this minimization can cause arbitrary increases in the third term, e.g. they can break down under shifting label distributions. We propose *asymmetrically-relaxed distribution alignment*, a new approach that overcomes some limitations of standard domain-adversarial algorithms. Moreover, we characterize precise assumptions under which our algorithm is theoretically principled and demonstrate empirical benefits on both synthetic and real datasets.

## 1 INTRODUCTION

Despite breakthroughs in supervised deep learning across a variety of challenging tasks, current techniques depend precariously on the i.i.d. assumption. Unfortunately, real-world settings often demand not just generalization to *unseen examples* but robustness under a variety of shocks to the data distribution. Ideally, our models would leverage unlabeled test data, adapting in real time to produce improved predictions. *Unsupervised domain adaptation* formalizes this problem as learning a classifier from labeled *source domain* data and unlabeled data from a *target domain*, to maximize performance on the target distribution.

Without further assumptions, guarantees of target-domain accuracy are impossible (Ben-David et al., 2010b). However, well-chosen assumptions can make possible algorithms with non-vacuous performance guarantees. For example, under the *covariate shift* assumption (Heckman, 1977; Shimodaira, 2000), although the input marginals can vary between source and target ($p_S(x) \neq p_T(x)$), the conditional distribution of the labels (given features) exhibits invariance across domains ($p_S(y|x) = p_T(y|x)$). Traditional approaches to the covariate shift problem require the source distributions' support to cover the target support, estimating adapted classifiers via importance-weighted risk minimization (Shimodaira, 2000; Huang et al., 2007; Gretton et al., 2009; Yu & Szepesvári, 2012; Lipton et al., 2018).

Problematically, assumptions of contained support are violated in practice. A recent sequence of deep learning papers have proposed empirically-justified adversarial training schemes aimed at practical problems with non-overlapping supports (Ganin et al., 2016; Tzeng et al.). Example problems include generalizing from gray-scale images to colored images or product images on white backgrounds to photos of products in natural settings. While importance-weighting solutions are useless here (with non-overlapping support, weights are unbounded), *domain-adversarial networks* (Ganin et al., 2016) and subsequently-proposed variants report strong empirical results on a variety of image recognition challenges.

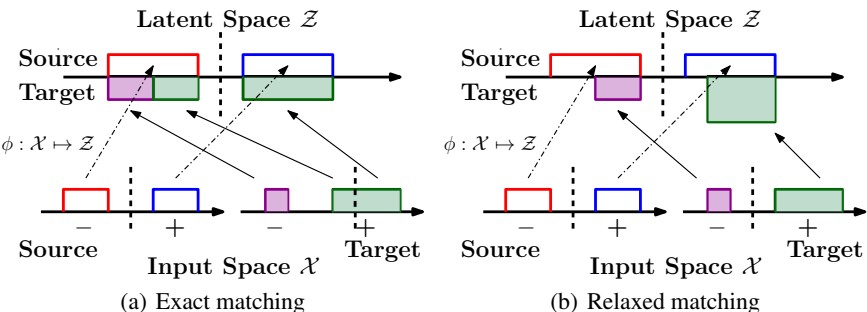

(a) Exact matching           (b) Relaxed matching

Figure 1: (a) In order to match the latent space distributions exactly, a model must map some elements of positive class in the target domain to some elements of negative class in the source domain. (b) A better mapping is achieved by requiring only that the source covers the target in the latent space.

The key idea of domain-adversarial networks is to simultaneously minimize the source error and align the two distributions in representation space. The scheme consists of an encoder, a *label classifier*, and a *domain classifier*. During training, the *domain classifier* is optimized to predict each image's domain given its encoding. The *label classifier* is optimized to predict labels from encodings (for source images). The encoder weights are optimized for the twin objectives of accurate label classification (of source data) and *fooling* the domain classifier (for all data).

Although Ganin et al. (2016) motivate their idea via theoretical results due to Ben-David et al. (2010a), the theory is insufficient to justify their method. Put simply, Ben-David et al. (2010a) bound the test error by a sum of three terms. The domain-adversarial objective minimizes two among these, but this minimization may cause the third term to increase. This is guaranteed to happen when the label distribution shifts between source and target (Figure 1(a)).

In this paper, we propose asymmetrically-relaxed distribution alignment, a relaxed distance for aligning data across domains that can be minimized without requiring latent-space distributions to match exactly. The new distance is minimized whenever the density ratios in representation space from target to source are upper bounded by a certain constant, such that the target representation support is contained in the source representation's. The relaxed distribution alignment need not lead to a poor classifier on the target domain under label distribution mismatch (Figure 1(b)). We demonstrate theoretically that the relaxed alignment is sufficient for a good target domain performance under a concrete set of assumptions on the data distributions. Further, we propose several practical ways to achieve the relaxed distribution alignment, translating the new distance into adversarial learning objectives. Empirical results on synthetic and real datasets show that incorporating our relaxed distribution alignment loss into adversarial domain adaptation gives better classification performance on the target domain. **Due to space constraints, we only briefly state our results in the main text and append the full version of our paper after references.**

## 2 BACKGROUND AND MOTIVATION

**Unsupervised domain adaptation with representations**     For simplicity, we address the binary classification scenario. Let $\mathcal{X}$ be the input space and $f : \mathcal{X} \mapsto \{0, 1\}$ be the (domain-invariant) ground truth labeling function. Let $p_S$ and $p_T$ be the input distributions over $\mathcal{X}$ for source and target domain respectively. Let $\mathcal{Z}$ be a latent space and $\Phi$ denote a class of mappings from $\mathcal{X}$ to $\mathcal{Z}$. Define $\mathcal{H}$ to be a class of predictors over the latent space $\mathcal{Z}$, i.e., each $h \in \mathcal{H}$ maps from $\mathcal{Z}$ to $\{0, 1\}$. Given a representation mapping $\phi \in \Phi$, classifier $h \in \mathcal{H}$, and input $x \in \mathcal{X}$, our prediction is $h(\phi(x))$. In the *unsupervised domain adaptation* setting, we have access to labeled source data $(x, f(x))$ for $x \sim p_S$ and unlabeled target data $x \sim p_T$. We are interested in bounding the classification risk of a $(\phi, h)$-pair on the target domain:

$$\mathcal{E}_T(\phi, h) = \mathcal{E}_S(\phi, h) + \int \mathrm{d}z p_T^\phi(z)\left(r_T(z; \phi, h) - r_S(z; \phi, h)\right) + \int \mathrm{d}z \left(p_T^\phi(z) - p_S^\phi(z)\right) r_S(z; \phi, h),$$

(1)

where $r$ is the risk function in the latent space.

**Domain-adversarial learning** *Domain-adversarial* approaches focus on minimizing the first and third term in (1) jointly. Informally, these approaches minimize the source domain classification risk and the distance between the two distributions in the latent space:

$$\min_{\phi, h} \mathcal{E}_S(\phi, h) + \lambda D(p_S^\phi, p_T^\phi) + \Omega(\phi, h) \,, \tag{2}$$

where $D$ is a distance metric between distributions and $\Omega$ is a regularization term. Standard choices of $D$ have the property that $D(p_S^\phi, p_T^\phi) = 0$ if $p_S^\phi \equiv p_T^\phi$ and $D(p_S^\phi, p_T^\phi) > 0$ otherwise. This exact distribution matching, however, can lead to undesirable performance. More specifically, the following proposition says that the target error is lower bounded if label distribution shifts:

**Proposition 2.1.** Let $\rho_S$ and $\rho_T$ be the proportion of data with positive label. If $D(p_S^\phi, p_T^\phi) = 0$ if and only if $p_S^\phi \equiv p_T^\phi$, $\mathcal{E}_S(\phi, h) = D(p_S^\phi, p_T^\phi) = 0$ indicates $\mathcal{E}_T(\phi, h) \geq |\rho_S - \rho_T|$.

This problem happens because although $D(p_S^\phi, p_T^\phi) = 0$ is a sufficient condition for the third term of (1) to be zero, *it is not a necessary condition.* We now examine the third term of (1): $\int \mathrm{d}z \left( p_T^\phi(z) - p_S^\phi(z) \right) r_S(z; \phi, h) \leq \left( \sup_{z \in \mathcal{Z}} \frac{p_T^\phi(z)}{p_S^\phi(z)} - 1 \right) \mathcal{E}_S(\phi, h)$. This expression shows that if the source error $\mathcal{E}_S(\phi, h)$ is zero then it is sufficient to say the third term of (1) is zero when the density ratio $p_T^\phi(z)/p_S^\phi(z)$ is upper bounded by some constant for all $z$, as shown in Figure 1(b).

Given this motivation, we propose relaxing from exact distribution matching to bounding the density ratio in the domain-adversarial learning objective (2). We call this *asymmetrically-relaxed distribution alignment.* More specifically, **our proposed approach** is to *replace the typical distribution distance $D$ in the domain-adversarial objective* (2) *with a $\beta$-admissible distance $D_\beta$ so that minimizing the new objective does not necessarily lead to a failure under label distribution shift.*

**Definition 2.2** ($\beta$-admissible distances). Given a family of distributions defined on the same space $\mathcal{Z}$, a distance metric $D_\beta$ between distributions is called $\beta$-*admissible* if $D_\beta(p, q) = 0$ when $\sup_{z \in \mathcal{Z}} p(z)/q(z) \leq 1 + \beta$ and $D_\beta(p, q) > 0$ otherwise.

## 3 THEORETICAL RESULTS

We bound the target domain error under our proposed asymmetrically-relaxed distribution alignment. Our theoretical result makes distinct contribution to the domain adaptation literature: We provide a risk bound that explains the behavior of domain-adversarial methods with *model-independent* assumptions on data distributions. Existing theories without assumptions of contained support (Ben-David et al., 2007; 2010a; Ben-David & Urner, 2014; Mansour et al., 2009; Cortes & Mohri, 2011) do not exhibit this property.

**Construction 3.1.** The following statements hold simultaneously: (1) (*Lipschitzness of representation mapping.*) $\phi$ is $L$-Lipschitz: $d_{\mathcal{Z}}(\phi(x_1), \phi(x_2)) \leq L d_{\mathcal{X}}(x_1, x_2)$ for any $x_1, x_2 \in \mathcal{X}$. (2) (*Imperfect asymmetrically-relaxed distribution alignment.*) For some $\beta \geq 0$, there exist a set $B \subset \mathcal{Z}$ such that $\frac{p_T^\phi(z)}{p_S^\phi(z)} \leq 1 + \beta$ holds for all $z \in B$ and $p_T^\phi(B) \geq 1 - \delta_1$. (3) (*Separation of source domain in the latent space.*) There exist two sets $C_0, C_1 \subset \mathcal{X}$ that satisfy: (a) $C_0 \cap C_1 = \emptyset$ (b) $p_S(C_0 \cup C_1) \geq 1 - \delta_2$. (c) For $i \in \{0, 1\}$, $f(x) = i$ for all $x \in C_i$. (d) $\inf_{z_0 \in \phi(C_0), z_1 \in \phi(C_1)} d_{\mathcal{Z}}(z_0, z_1) \geq \Delta > 0$.

**Assumption 3.2.** (*Connectedness from target domain to source domain.*) Given constants $(L, \beta, \Delta, \delta_1, \delta_2, \delta_3)$, assume that, for any $B_S, B_T \subset \mathcal{X}$ with $p_S(B_S) \geq 1 - \delta_2$ and $p_T(B_T) \geq 1 - \delta_1 - (1+\beta)\delta_2$, there exists $C_T \subset B_T$ that satisfies the following conditions: (1) For any $x \in C_T$, there exists $x' \in C_T \cap B_S$ such that one can find a sequence of points $x_0, x_1, ..., x_m \in C_T$ with $x_0 = x, x_m = x', f(x) = f(x')$ and $d_{\mathcal{X}}(x_{i-1}, x_i) < \frac{\Delta}{L}$ for all $i = 1, ..., m$. (2) $p_T(C_T) \geq 1 - \delta_3$.

**Theorem 3.3.** Given a $L$-Lipschitz mapping $\phi \in \Phi$ and a binary classifier $h \in \mathcal{H}$, if $\phi$ satisfies the properties in Construction 3.1 with constants $(L, \beta, \Delta, \delta_1, \delta_2)$, and Assumption 3.2 holds with the same set of constants plus $\delta_3$, then the target domain error can be bounded as

$$\mathcal{E}_T(\phi, h) \leq (1 + \beta)\mathcal{E}_S(\phi, h) + 3\delta_1 + 2(1 + \beta)\delta_2 + \delta_3 \,.$$

## 4 ASYMMETRICALLY-RELAXED DISTANCES

In this section, we derive several $\beta$-*admissible* distance metrics that can be practically minimized with adversarial training.

$f$**-divergence**   We propose a general approach to make any $f$-divergence $\beta$-admissible by partially linearizing the function $f$. Plugging in the corresponding $f$ for JS-divergence gives

$$D_{\bar{f}_\beta}(p, q) = \sup_{g:\mathcal{Z}\mapsto(0,1]} \mathbb{E}_{z\sim q}\left[\log\frac{g(z)}{2+\beta}\right] + \mathbb{E}_{z\sim p}\left[\log\left(1 - \frac{g(z)}{2+\beta}\right)\right] . \qquad (3)$$

**Wasserstein distance**   The idea behind modifying the Wasserstein distance is to model the optimal transport from $p$ to the region where distributions have $1 + \beta$ maximal density ratio with respect to $q$. Following the dual-form derivation for the original Wasserstein distance gives

$$W_\beta(p, q) = \sup_g \mathbb{E}_{z\sim p}\left[g(z)\right] - (1+\beta)\mathbb{E}_{z\sim q}\left[g(z)\right] \qquad (4)$$

$$\text{s.t. } \forall z \in \mathcal{Z}, g(z) \geq 0, \forall z_1, z_2 \in \mathcal{Z}, g(z_1) - g(z_2) \leq \|z_1 - z_2\| ,$$

**Reweighting distance**   Given any distance metric $D$, a generic way to make it $\beta$-admissible is to allow reweighting for one of the distances within a $\beta$-dependent range: Given a distribution $q$ over $\mathcal{Z}$ and a reweighting function $w : \mathcal{Z} \mapsto [0, \infty)$. The reweighted distribution $q_w$ is defined as $q_w(z) = \frac{q(z)w(z)}{\int \mathrm{d}zq(z)w(z)}$. Define $\mathcal{W}_{\beta,q}$ to be a set of $\beta$-*qualified* reweighting with respect to $q$: $\mathcal{W}_{\beta,q} = \left\{w : \mathcal{Z} \mapsto [0,1], \int \mathrm{d}zq(z)w(z) = \frac{1}{1+\beta}\right\}$. Then the relaxed distance can be defined as $D_\beta(p, q) = \min_{w\in\mathcal{W}_{\beta,q}} D(p, q_w)$.

## 5 EXPERIMENTS

To evaluate our approach, we implement Domain Adversarial Neural Networks (DANN), (Ganin et al., 2016) replacing the JS-divergence with several $\beta$-admissible distances. Table 5-2 summarize our experimental results. Compared to the original DANN, our approaches fare significantly better under label distribution shift while achieving comparable performance absent label distribution shift.

| METHOD | ACCURACY% | | |
|---|---|---|---|
| SOURCE | 89.4±1.1 | | |
| DANN | 59.1±5.1 | WDANN | 50.8±32.1 |
| $\beta$ | 0.5 | 2.0 | 4.0 |
| FDANN-$\beta$ | 66.0± 41.6 | **99.9± 0.0** | 99.8±0.0 |
| sDANN-$\beta$ | **99.9± 0.1** | **99.9± 0.0** | 99.9±0.0 |
| WDANN1-$\beta$ | 45.7± 41.5 | 66.4± 41.1 | 99.9±0.0 |
| WDANN2-$\beta$ | 97.6± 1.2 | 99.7± 0.2 | 99.5±0.3 |
| sWDANN-$\beta$ | 79.0± 5.9 | **99.9± 0.0** | **99.9±0.0** |

Table 1: Classification accuracy on target domain with label distribution shift on a synthetic dataset.

| TARGET LABELS | [0-4] SHIFT | [5-9] SHIFT | [0-9] NO-SHIFT | TARGET LABELS | [0-4] SHIFT | [5-9] SHIFT | [0-9] NO-SHIFT |
|---|---|---|---|---|---|---|---|
| SOURCE | 74.3±1.0 | 59.5±3.0 | 66.7±2.1 | SOURCE | 69.4±2.3 | 30.3±2.8 | 49.4±2.1 |
| DANN | 50.0±1.9 | 28.2±2.8 | 78.5±1.6 | DANN | 57.6±1.1 | 37.1±3.5 | **81.9±6.7** |
| FDANN-1 | 71.6±4.0 | **67.5±2.3** | 73.7±1.5 | FDANN-1 | 80.4±2.0 | 40.1±3.2 | 75.4±4.5 |
| FDANN-2 | 74.3±2.5 | 61.9±2.9 | 72.6±0.9 | FDANN-2 | **86.6±4.9** | 41.7±6.6 | 70.0±3.3 |
| FDANN-4 | 75.9±1.6 | 64.4±3.6 | 72.3±1.2 | FDANN-4 | 77.6±6.8 | 34.7±7.1 | 58.5±2.2 |
| SDANN-1 | 71.6±3.7 | 49.1±6.3 | 81.0±1.3 | SDANN-1 | 68.2±2.7 | **45.4±7.1** | 78.8±5.3 |
| SDANN-2 | 76.4±3.1 | 48.7±9.0 | 81.7±1.4 | SDANN-2 | 78.6±3.6 | 36.1±5.2 | 77.4±5.7 |
| SDANN-4 | **81.0±1.6** | 60.8±7.5 | **82.0±0.4** | SDANN-4 | 83.5±2.7 | 41.1±6.6 | 75.6±6.9 |

Table 2: Classification accuracy on target domain with/without label distribution shift on MNIST-USPS.

Table 3: Classification accuracy on target domain with/without label distribution shift on USPS-MNIST.

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
