# OpenReview forum: "Domain Adaptation with Asymmetrically-Relaxed Distribution Alignment"
_ICLR.cc/2019/Workshop/LLD — LLD 2019_

### Official Review · AnonReviewer1 · 2019-04-07
**Attached paper is good but short version needs rework**

**Rating:** 4
**Confidence:** 2

**Review:**

This paper identifies a limit of the theoretical framework of Ben-David et al. (2010), regarding an upper bound of the target error in the domain adaptation setting, which is the sum of 3 terms. Authors show that Domain adversarial approach from Ganin et al. (2016) focuses on minimizing 2 of these 3 terms but leaving out the third term leads to a lower bound on the target error, which increases when the difference between source and target distributions augment. The paper then suggests to used relaxed metrics to prevent this effect : giving several examples of such metrics, the authors propose new theoretical bounds on the target error.

The attached paper (12 pages, excluding proofs) is very clear and interesting. The framework and the theorems are stated clearly and, despite the technicality of the theorems, the key ideas seem easy to follow. The short (workshop) paper, however, is not as pleasant to read: the authors clearly lacked space, which especially shows in the last section, where no comment or explanations of the experiment are provided. Depending on the LLD organizers, this may be a problem. I suggest that the authors drop a few more results and propositions, for example concentrating on the β-f-divergences and the experiments on MNIST-USPS, which seem to be convincing enough.

Several points could also be detailed:
1. Why are WDANN variants absent from the final experiments?
2. The new upper bound on target error involves several terms, provided by assumptions on the source and target domains; is there a practical advantage of this formulation? How do these smaller bounds vary when the source and target distributions shift? This seems to be a central point in advocating the relevance of the proposed approach, which could benefit the longer paper.
3. The connected assumption 3.2 is intriguing. Does it appear in related works? Did you find datasets from which it is absent, or which provide a bad δ₃ bound?

---

### Official Review · AnonReviewer2 · 2019-04-10
**Interesting method**

**Rating:** 3
**Confidence:** 2

**Review:**

The paper proposes an asymmetrically-relaxed distribution alignment approach, to do unsupervised domain adaptation. For this, they propose 3 different "relaxed" distances.

Pros:
- The paper is well written and, although dense, quite clear.
- The proposed models are a good alternative to the original DANN
- The long version of the paper could be submitted to a journal.

Cons:
- The paper is very dense.
- Experiments section is very short and explanation of results is minimal. We do not know what are the different acronyms because they are not defined. They are only defined in the long version of the paper, that was attached.
- Conclusions are lacking.

---

### Decision · Program_Chairs · 2019-04-16
**Acceptance Decision**

Accept